

# Bio-Rollup: a new privacy protection solution for biometrics based on two-layer scalability-focused blockchain

Jian Yun[*], Yusheng Lu[*], Xinyang Liu and Jingdan Guan

Dalian Minzu University, College of Computer Science and Engineering, Dalian, Liaoning, China
[*] These authors contributed equally to this work.

## ABSTRACT

The increased use of artificial intelligence generated content (AIGC) among vast user populations has heightened the risk of private data leaks. Effective auditing and regulation remain challenging, further compounding the risks associated with the leaks involving model parameters and user data. Blockchain technology, renowned for its decentralized consensus mechanism and tamper-resistant properties, is emerging as an ideal tool for documenting, auditing, and analyzing the behaviors of all stakeholders in machine learning as a service (MLaaS). This study centers on biometric recognition systems, addressing pressing privacy and security concerns through innovative endeavors. We conducted experiments to analyze six distinct deep neural networks, leveraging a dataset quality metric grounded in the query output space to quantify the value of the transfer datasets. This analysis revealed the impact of imbalanced datasets on training accuracy, thereby bolstering the system's capacity to detect model data thefts. Furthermore, we designed and implemented a novel Bio-Rollup scheme, seamlessly integrating technologies such as certificate authority, blockchain layer two scaling, and zero-knowledge proofs. This innovative scheme facilitates lightweight auditing through Merkle proofs, enhancing efficiency while minimizing blockchain storage requirements. Compared to the baseline approach, Bio-Rollup restores the integrity of the biometric system and simplifies deployment procedures. It effectively prevents unauthorized use through certificate authorization and zero-knowledge proofs, thus safeguarding user privacy and offering a passive defense against model stealing attacks.

# INTRODUCTION

Biometric recognition technology has been widely adopted since the 1970s. However, data security and privacy protection for biometric recognition systems is still unreliable. Biometric information is strongly correlated with an individual's physiological characteristics or behaviors. Therefore, biometric template protection technology is a critical research area in data security and privacy protection for biometric recognition systems. Numerous biometric template protection schemes have been proposed in this area, such as methods based on irreversible and reversible transformations of features (*Ratha, Connell & Bolle, 2001*; *Jin, Ling & Goh, 2004*) and methods based on biometric encryption

Corresponding author
Jian Yun, yunjianm@163.com

for key generation and binding of templates (*Nichols, 1998*; *Monrose, Reiter & Wetzel, 1999*).

*Tramèr et al. (2016)* demonstrated the potential for privacy leaks in machine learning models. As a typical application of artificial intelligence, biometric recognition systems are inevitably exposed to the threat of privacy attacks. Numerous technical solutions aim to safeguard biometric recognition systems' security and privacy. However, centralized storage and deployment of these systems remain vulnerable to attacks. The security vulnerabilities of biometric recognition systems are illustrated in the Fig. 1.

Blockchain technology addresses the various challenges that biometric recognition systems face by ensuring data security and privacy, enabling reliable digital identity management, supporting smart contracts, and providing transparent and auditable operations. The integration of blockchain technology with biometric recognition systems offers the following advantages:

- Data security and privacy protection: The decentralized, immutable, and transparent nature of blockchain technology provides practical technical support for the secure storage and privacy protection of biometric data.
- Authentication and authorization mechanisms: Blockchain technology facilitates verifiable and decentralized digital identity management, offering secure authentication and authorization mechanisms for biometric recognition technologies.
- Enhanced reliability: By introducing distributed ledgers, blockchain ensures the reliability and traceability of data within biometric recognition systems.
- Transparency and audibility: The transparency and audibility of blockchain technology provide effective monitoring of biometric recognition system operations, thereby enhancing the overall trustworthiness of the system.

The convergence of blockchain and biometric recognition technologies collectively presents a robust solution for advancing secure and trustworthy identity verification processes. Since 2018, research on the integration of blockchain technology with artificial intelligence (AI) (*Zheng, Dai & Wu, 2019*; *Salah et al., 2019*), machine learning (ML) (*Liu et al., 2020a*; *Chen et al., 2018*), and biometric recognition systems has gained widespread attention (*Huang et al., 2021*). There have been new developments in the integration of biometric recognition systems and blockchain technology. *Goel et al. (2019)* combined biometric recognition feature extraction and template matching with blockchain, utilizing no tray blocks in the feature extraction process to achieve consensus in the blockchain, and using Shamir key sharing threshold schemes to vote on matching results to restore keys in the template matching stage. Despite addressing security concerns in biometric recognition systems, such as machine learning models, feature extraction data, and user biometric templates, existing solutions have notable shortcomings:

- Given the current limitations in blockchain efficiency and scalability, the usability and versatility of these systems should be enhanced.
- The absence of auditing and tracking systems for unauthorized access leaves the systems vulnerable to privacy attacks.

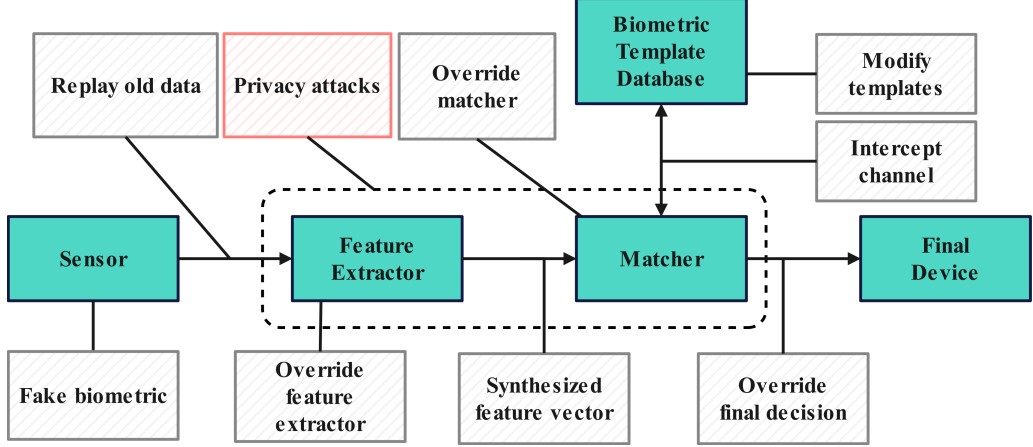

**Figure 1** Weaknesses in biometrics systems.

- Insufficient definitions and explanations of user data privacy fail to address the privacy challenges posed by blockchain data transparency.

Against this backdrop, this study aims to bridge the gap in research on decentralized AI security frameworks leveraging blockchain as machine learning as a service enters a period of rapid development and deployment. The research content is significant for advancing the field of AI security frameworks.

## Contributions

In the context of the burgeoning AIGC sector, the deployment of models across a broad user base has amplified the vulnerability to privacy data breaches. Blockchain technology, characterized by its decentralized consensus and immutability, stands out as a potent tool for logging, auditing, and scrutinizing the actions of all parties within MLaaS. Nevertheless, existing literature has not delved extensively into the performance scalability of blockchain or the defense against AI privacy attacks.

Given this gap, this study concentrates on the privacy security of biometric recognition systems, introducing an innovative AI privacy protection scheme, Bio-Rollup, which amalgamates blockchain layer-two scaling and biometric identification. The core concept of Bio-Rollup is to integrate the deployment and query of biometric recognition systems within the auditing scope of blockchain layer-two scaling nodes, thereby effectively validating the legitimacy of model deployment, modifications, and user queries. The contributions of this study include:

- Experimental analysis of six distinct deep neural networks, employing a dataset quality metric rooted in query output space to quantify the value of migrated datasets and elucidate the impact of imbalanced datasets on training accuracy, thereby bolstering the system's capacity to detect model stealing attacks.
- Design and implementation of the Bio-Rollup scheme, which integrates certificate authorization, blockchain layer-2 scaling, and zero-knowledge proofs. The scheme

**Table 1  Acronym description.**

| Acronym | Explanation |
| --- | --- |
| AI | Artificial Intelligence |
| ML | Machine Learning |
| MLaaS | Machine Learning as a Service |
| IoT | Internet of Things |
| BaaS | Blockchain as a Service |
| P2P | Network peer-to-peer Network |
| CA | Certificate Authority |
| Dapp | Distributed Application |
| AIGC | Artificial Intelligence Generated Content |
| SNARK | Succinct Non-interactive Argument of Knowledge |
| GAM | Generalized Additive Model |

achieves lightweight auditing through Merkle proofs, enhancing efficiency and alleviating blockchain storage pressure. In contrast to baseline approaches, Bio-Rollup restores the integrity of the biometric system, reduces deployment complexity, and prevents unauthorized use through certificate authorization and zero-knowledge proofs, ensuring user privacy and offering passive defense against model stealing attacks.

In conclusion, this study offers robust privacy protection for user data within biometric recognition systems while ensuring efficiency, usability, and security. It also charts new research directions in the integration of blockchain technology with biometric recognition.

To enhance readability, a list of acronyms is provided in Table 1. The structure of this article is as follows: Related work is discussed in "Literature Review". "Preliminary Knowledge" introduces the key concepts necessary to understand the rest of this article, such as blockchain and ZK-snarks. The methodology employed in our framework is detailed in "Proposed Architecture". In the "Experimental Evaluation and Discussion", the idea and effectiveness of the scheme are further demonstrated. Finally, "Conclusions" brings the article to a close and outlines future work.

## LITERATURE REVIEW

Biometric recognition technology aims to automatically verify user identity by using physiological features such as facial or fingerprint patterns, or behavioral features such as voice or handwritten signatures (*Jain et al., 2016*). AI systems face different security risks at various stages of their lifecycle (*Liu, 2020*). Biometric recognition systems are a typical use of AI that are mainly characterized by insecure hardware and software environments during deployment. These challenges may lead to security risks such as unauthorized machine learning model modification and unauthorized access by non-authorized users (*Jing et al., 2021*). These problems are typically manifested in model extraction attacks.

## Blockchain and biometrics

Since 2018, the integration of blockchain technology with AI (*Salah et al., 2019*), ML (*Chen et al., 2018*; *Liu et al., 2020b*), and biometric recognition systems has garnered significant attention (*Huang et al., 2021*).

The integration of blockchain with biometric recognition systems is an emerging and innovative field. The current research primarily focuses on literature reviews that analyze application domains, advantages, and legal implications. For instance, *Delgado-Mohatar et al. (2020)* delved into the application of blockchain technology for biometric template storage and protection, evaluating the benefits and challenges of various blockchain architectures within biometric systems. The research highlighted the potential of Merkle tree structures for cost reduction and quantified the expenses of both off-chain and on-chain matching, demonstrating the enhancing role of template protection in matching accuracy. *Ghafourian et al. (2023)* provided an overview of the potential benefits and risks of merging blockchain with biometric recognition, along with a synthesis of technical aspects and an inaugural legal analysis. The synthesis underscored the significant promise for innovative applications in the biometric domain, despite the integration being in its infancy. The allocation of liability remains a paramount legal concern, along with other challenges such as conducting proper data protection impact assessments. *Sharma & Dwivedi (2024)* offered a comprehensive review of blockchain's application in biometric systems, emphasizing its potential to enhance security, transparency, and traceability. The survey outlined the fundamental principles and challenges of blockchain technology, as well as the pivotal operations of biometric systems, exploring application areas such as template storage, identity management, and authentication.

These reviews collectively provide a foundational understanding of the integration of blockchain with biometric systems, charting a course for future research directions. Additionally, there are scholarly works that propose specific solutions for targeted scenarios. For instance, *Goel et al. (2019)* introduced an integrated architecture that enhanced the security of biometric systems by employing blockchain's immutability to encrypt and verify the feature extraction process and employing a Merkle tree-like structure for secure, decentralized matching. *Alharthi, Ni & Jiang (2021)* presented a BBC-based privacy protection framework that ensures secure and trustworthy vehicle communications in VANETs while safeguarding user privacy. The framework uses cancellable biometric information to create unique pseudonyms, maintaining privacy while allowing traceability. *Lee & Jeong (2021)* introduced the blockchain-based biometric authentication system (BDAS), which enhanced the security, reliability, and transparency of biometric authentication through distributed management of biometric information. BDAS fragments biometric templates and employs blockchain for decentralized management, establishing a decentralized authentication mechanism.

In conclusion, integrating blockchain technology into biometric recognition systems represents a novel research direction. To date, no studies have proposed the use of blockchain's authentication and authorization mechanisms and auditing capabilities to achieve comprehensive privacy protection across the entire lifecycle of biometric recognition systems. Considering this *Goel et al. (2019)* was used as a baseline for

comparative analysis. Firstly, the proposed baseline scheme mapped the feature extraction process onto the blockchain, with each extraction step corresponding to a block. The integrity of the feature extraction process was verified through the introduction of notarized blocks. While this design enhances audit precision, it sacrifices feature extraction efficiency and increases the complexity of deploying the biometric recognition system. This study suggests that integrity recovery for biometric systems should be prioritized, with user queries processed as batched transactions for auditing. Secondly, we built a blockchain-based authentication and authorization mechanism on a consortium chain to prevent unauthorized use and provide audit conditions. Finally, by analyzing the user's output space, the study assesses the extent to which users can access private data and designs an artificial intelligence security framework, Bio-Rollup, based on a second-layer scalability off-chain auditing mechanism to safeguard the privacy of biometric recognition systems. Experimental results indicate that the proposed study's scheme, employing two zero-knowledge proof protocols, significantly improves audit efficiency compared to the baseline and effectively warns against privacy attacks involving data leaks.

## Privacy attacks

Model stealing attacks represent a clandestine form of black-box assaults where the attacker's fundamental objective is to construct a substitute model $f'$ that replicates the functionality and performance of the victim model $f$. The primary purposes of such attacks can be encapsulated in two broad aspects: firstly, to ensure that $f'$ achieves a comparable level of accuracy to $f$ on a test set drawn from the input data distribution that is pertinent to the learning task (*Krishna et al., 2019*; *Milli et al., 2019*; *Orekondy, Schiele & Fritz, 2019*; *Tramèr et al., 2016*); secondly, to create an $f'$ that closely aligns with $f$ on a set of input points that have no direct correlation with the learning task (*Correia-Silva et al., 2018*; *Jagielski et al., 2020*; *Juuti et al., 2019*; *Tramèr et al., 2016*). *Jagielski et al. (2020)* termed the former type of attack as "task accuracy extraction", while the latter was referred to as "loyalty extraction". Model stealing attacks can be leveraged to launch subsequent assaults of various types, including adversarial attacks (*Juuti et al., 2019*; *Papernot et al., 2017*) and membership inference attacks (*Nasr, Shokri & Houmansadr, 2019*). Furthermore, there are studies focused on extracting information from the victim model, such as extracting hyperparameters from the target function (*Wang & Gong, 2018*) or various attributes of neural network architectures, including activation types, optimization algorithms, number of layers, and more (*Oh, Schiele & Fritz, 2019*). A summary of model stealing attacks is provided in Table 2.

Passive defense strategies focus on detecting malicious querying behavior and limiting or denying inference services to attackers. *Kesarwani et al. (2018)* developed an extraction auditing mechanism tailored for decision tree models. *Juuti et al. (2019)* introduced PRADA, a multi-query detection approach. *Kariyappa & Qureshi (2020)* proposed an adaptive error information injection method that selectively sends incorrect predictions for queries deemed to be out-of-distribution. However, these countermeasures have their limitations; for instance, active defense strategies may compromise model accuracy, while passive defense strategies incur high computational costs when dealing with large-scale

**Table 2   Summary of model stealing attacks.**

| Related work | Adversary knowledge | Model | Attack phase |
|---|---|---|---|
| *Tramèr et al. (2016)* | Blackbox/Whitebox | Neural network, *etc* | Inference |
| *Papernot et al. (2017)* | Blackbox | Neural network | Inference |
| *Correia-Silva et al. (2018)* | Blackbox | Neural network | Inference |
| *Oh, Schiele & Fritz (2019)* | Blackbox | Neural network | Inference |
| *Juuti et al. (2019)* | Blackbox | Neural network | Inference |
| *Milli et al. (2019)* | Blackbox | Neural network | Inference |
| *Orekondy, Schiele & Fritz (2019)* | Blackbox | Neural network | Inference |
| *Barbalau et al. (2020)* | Blackbox | Neural network | Inference |
| *Chandrasekaran et al. (2020)* | Blackbox | Neural network | Inference |
| *Jagielski et al. (2020)* | Blackbox | Neural network | Inference |
| *Pal et al. (2020)* | Blackbox | Neural network | Inference |
| *Yu et al. (2020)* | Blackbox | Neural network | Inference |
| *Gong et al. (2020)* | Blackbox | Neural network | Inference |

**Table 3   Comparison of defense strategies against model stealing attacks.**

| Related work | Privacy | Tamper resistance | Attack knowledge | Defense strategy |
|---|---|---|---|---|
| *Juuti et al. (2019)* | – | – | Query, Model | Detection |
| *Kariyappa & Qureshi (2020)* | – | – | Query, Gradient | Detection |
| *Kesarwani et al. (2018)* | – | – | Query | Detection |
| This scheme | ✓ | ✓ | Query | Blockchain, Detection |

data, and some methods are only applicable to specific types of models. Consequently, future research needs to enhance the generality and effectiveness of defense strategies without compromising model performance. A comparison of this study with existing approaches is provided in Table 3.

# PRELIMINARY KNOWLEDGE

## Blockchain

Blockchain is a distributed ledger that relies heavily on hash functions and cryptography to store redundant data. It enables peer-to-peer transactions, coordination, and collaboration in a decentralized system without central institutions. This is achieved through various techniques such as data encryption, timestamping, distributed consensus, and economic incentives (*Yuan & Wang, 2016*).

Smart contracts (*Zheng et al., 2020*) are computer programs that encode the terms of a contract into code that can be executed on a blockchain. Each contract is created and executed as a transaction, which is recorded on the blockchain. When a smart contract is called, it is executed by all nodes in the network, ensuring that the contract remains active even if one node fails. The most popular public chain for smart contracts is Ethereum (*Buterin, 2015*), which supports programming languages such as Solidity and

Vyper (*Kaleem, Mavridou & Laszka, 2020*). Hyperledger Fabric (a consortium chain) uses container technology to host smart contract codes.

The scaling technology of blockchain layer two (*Gangwal, Gangavalli & Thirupathi, 2022*) aims to balance the need for decentralization and security with the requirement for efficient transaction processing. State channels (*Dziembowski, Faust & Hostáková, 2018*) and rollups are currently popular L2 solutions. State channels utilize multi-signature smart contracts to enable users to make transactions offline before final settlement on the main net, providing free transactions. Rollups execute transactions off-chain before compressing and publishing the original data on the main net. Two types of rollups are: Optimistic rollups, which propose fraud proofs to ensure the correctness of the off-chain state (*Optimism, 2021*; *Boba, 2021*), while ZK-Rollups upload zero-knowledge proofs to guarantee the accuracy of the off-chain state (*Matter-Labs, 2019*; *Aztec, 2020*; *Starkware, 2020*).

### Succinct non-interactive argument of knowledge

Zero-knowledge proof is a technique that allows one party to prove the accuracy of a statement to another party without revealing any additional information beyond what is necessary (*Goldwasser & Micali, 1989*). It is characterized by completeness, reliability, and zero knowledge. The succinct non-interactive argument of knowledge (SNARKs) is a concise and non-interactive version of zero-knowledge proofs. SNARKs are achieved through the use of common reference strings (CRS) models (*Blum, Feldman & Micali, 2019*) and random oracle models (ROM) (*Bellare & Rogaway, 1993*). The statement to be proven can be converted into a circuit satisfiability problem (C-SAT), and the witness generated from the process data in the circuit is called a constraint. Groth16 is a general non-interactive zero-knowledge proof scheme based on the protocols of QAP (*Gennaro et al., 2013*) and LIP (*Bitansky et al., 2013*). It has linear proof size and constant verification time but requires a specific trusted setup for the pre-processing phase. GKMMM18 (*Groth et al., 2018*), Sonic (*Maller et al., 2019*), and PlonK (*Gabizon, Williamson & Ciobotaru, 2019*) implement global, updateable CRSs to address the trusted setup issue. PlonK generates more extensive proof data but has a faster generation speed than Groth16. Therefore, Groth16 is more suitable for applications that require generating many proofs with high-performance requirements. At the same time, PlonK is more suitable for scenarios where different circuits need to be processed to avoid the additional performance overhead caused by a trusted setup.

In the realm of lightweight zero-knowledge proofs, recent advancements have introduced a privacy-preserving authentication system for IoT applications (*Dwivedi et al., 2022*), which leverages non-interactive zero-knowledge proofs. This system features the lightweight ZKNimble encryption algorithm, allowing for authentication to be completed without users having to disclose any personal identification information.

## PROPOSED FRAMEWORK

### Threat model

The threat model encompasses the following categories of participants:

- Data owners: Data owners possess training datasets or other raw data that may contain sensitive information. Consequently, the regulation and auditing of data owners' actions are crucial for ensuring the confidentiality and integrity of the data.
- Model owners: Model owners have the right to access the data owned by data owners and are willing to share information related to their models. The regulation of model owners' behavior is essential for preventing unauthorized data access and model misuse.
- Cloud service providers: Cloud service providers are responsible for processing user query requests and managing the responses provided by the models. The regulation of cloud service providers' actions is critical for ensuring the security and compliance of data processing.
- Model users: Model users typically utilize the services provided by model owners through applications or user interfaces. The regulation of model users' behavior is significant for preventing the misuse of model services. Audits can ensure that model users adhere to usage terms and take appropriate security measures to protect their data.
- Adversaries: Adversaries may access the model's interface as normal model users do, and, if permitted, may directly access the model itself. Within the threat model, regulating the behavior of adversaries is vital for preventing privacy attacks. Audits can ensure that data owners comply with data protection regulations, such as GDPR, and take appropriate security measures to safeguard data. They can also ensure that model owners adhere to privacy regulations and implement suitable security measures to protect models and user data. Furthermore, audits can verify that cloud service providers comply with privacy regulations and employ appropriate security measures to protect data. Audits can detect abnormal behaviors by adversaries, such as frequent inquiries or attempts to obtain sensitive information and prompt the implementation of appropriate security measures to protect data. In summary, regulation and behavioral audits are imperative for ensuring data security and privacy protection. Through audits, potential security threats can be detected and prevented, and appropriate security measures can be taken to protect data.

## Assumptions

The following are several important assumptions related to experiments on the issue of privacy leakage in biometric recognition systems:

- Dataset imbalance: This study assumes that the remaining portion of the training set $1-t$ is evenly distributed across categories, partitioning the dataset into $t$ and $1-t$ segments, where t represents the proportion of the dataset classified as user biometric templates. The data set balance analysis experiment aims to validate that higher user loyalty leads to a stronger demand for self-related information, resulting in imbalanced sample collection and reduced model accuracy. Conversely, a higher degree of balance in the user queries indicates that attempting to steal the model may infringe upon privacy.
- Unauthorized access: The unauthorized use of biometric recognition systems encompasses malicious model deployment and privacy breaches across multiple dimensions. Unauthorized model owners may deploy malicious models, potentially collecting and transmitting biometric data without user knowledge or erroneously

associating user biometric information, thereby infringing privacy rights. Additionally, malicious access by unauthorized users may lead to privacy leaks, with attackers gaining system access through social engineering, physical attacks, or technical means to steal or abuse biometric data. Insiders also have the potential to abuse their privileges to disclose sensitive information. To address these issues, it is necessary to implement secure model deployment, enhance access control, auditing, and monitoring measures to mitigate unauthorized use risks and safeguard user privacy and data security. Performance analysis experiments demonstrate the efficiency advantage of the authorization and auditing scheme based on this proposal over the baseline scheme.

- Account and permission abuse: Biometric recognition systems authenticate individuals by identifying and verifying their biometric traits, which includes fingerprints, facial recognition, iris scans, palmprints, and voiceprints. Despite offering high security and convenience, these systems have vulnerabilities that could lead to account hijackings, such as attackers deceiving the system with impersonation techniques or copying biometric data during registration and transmission. Software vulnerabilities may be exploited by attackers. Furthermore, attackers could mimic users through replay attacks or steal biometric data through database leaks. Social engineering attacks may deceive users into sharing biometric data. To mitigate the risk of account hijacking, strict access control and monitoring auditing measures must be implemented to enhance system security and protect user identity privacy. Privacy attack prevention experiments reveal anomaly analysis of legitimate accounts engaging in privacy attacks after misuse.

## Proposed architecture

Bio-Rollup is composed of four parts: certificate authority (CA), biometric recognition system, blockchain layer two scaling service, and consortium chain, as shown in Fig. 2.

The CA is responsible for managing the organization and identity within the scheme, generating identity certificates for system administrators and users, and providing conditions for tracking the use behavior of the biometric recognition system. By setting up the CA, the layer two scaling service, biometric recognition system, and account are organized based on certificate authorization. This system uses communication certificates to build communication channels, complete end-to-end encryption, and secure transmission of user biometrics, effectively preventing users' biometrics from being tampered with or forged. At the same time, certificates can be revoked in a timely manner when suspected privacy attacks are detected based on the identity certificate authentication mechanism.

To address the issue of existing schemes using deconstructed deep neural networks for system deployment, which may result in model structure and parameter leakage due to improper block configuration file storage. Bio-Rollup's biometric recognition system uses Apache-TVM for deployment. To prevent tampering, the deployment of models serves as the initial transaction of the entire system, and any modifications to models also need to be reviewed by the system. The system state will iterate based on this initial transaction.

The layer two scaling service of Bio-Rollup is a decentralized application leveraging zero-knowledge proof (ZKP) functions to audit and verify the legality of system usage behavior. Bio-Rollup associates biometric data with user accounts and system behavior, facilitating

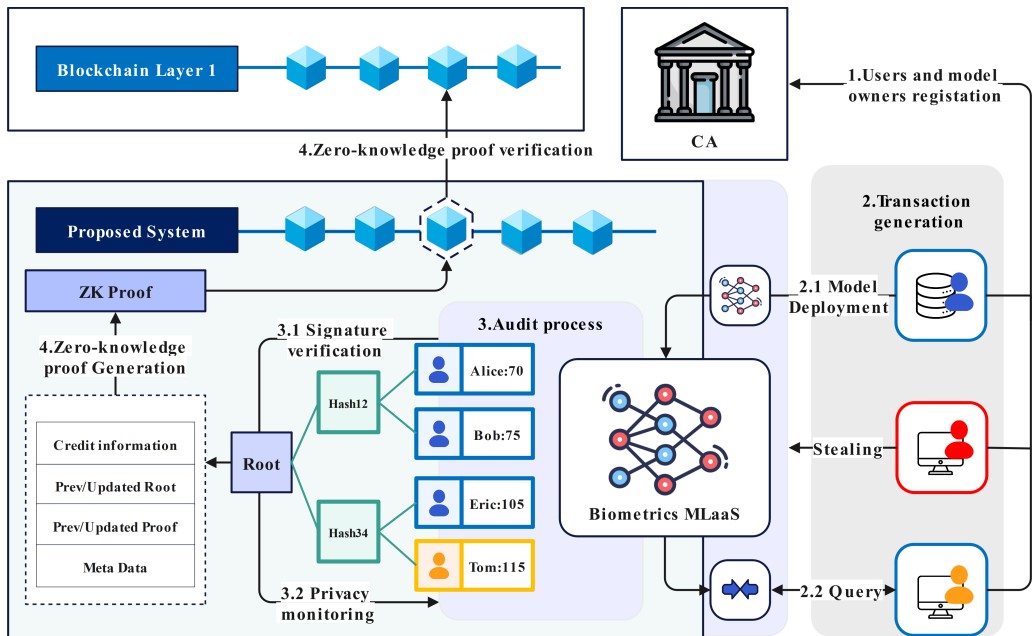

**Figure 2** **Overall structure of Bio-Rollup.** Image credit: Yusheng Lu, The source for the components at 10.5281/zenodo.12195012.

traceability. It distinguishes between manager and visitor accounts, with information stored in Merkle trees encompassing privacy data hash value $D_p$, credit parameter $C$, attack probability $\xi$, public key $P_k$, and sequence number $N$. This binding enables tracking and verification of authorized usage through CA identity management and Merkle audit paths.

## Workflow of proposed architecture

The overall workflow of the proposed scheme is shown in Fig. 3. Here, we provide a detailed overview of the process of the proposed scheme, combining Figs. 2 and 3.

Step 1: Participant registration phase, where all participants complete the registration process. This stage serves as the cornerstone of the entire procedure, ensuring the legitimacy and authenticity of the participants, and laying a solid foundation for subsequent transaction and verification processes.

Step 2: Transaction signature verification phase, which involves authenticating the signatures of transactions to ensure their authenticity and validity. This step is crucial for securing the integrity of transactions.

Step 3: Updating the Bio-Rollup status, which may include the updating of biometric information or the rolling status, thereby enhancing the system's security.

Step 4: Setting of the post-update status as witness, where the state change information is recorded and preserved, providing a reliable basis for subsequent audits.

Step 5: Initialization of user transactions, where user transactions are prepared for later signature verification and the generation of zero-knowledge proofs.

Step 6: Entailing the system checking the legality of the signatures and the authenticity of the data owners to ensure that transactions meet the predefined conditions.

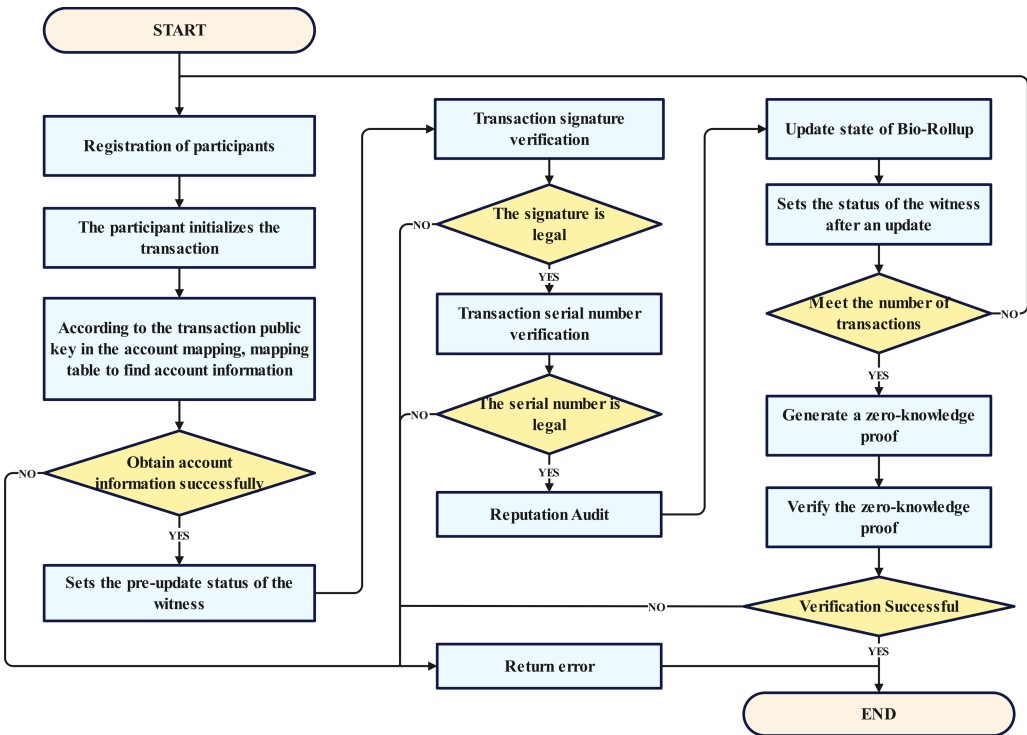

**Figure 3** General workflow.

Step 7: If the conditions are met, the seventh step involves finding the account within the user quantity mapping based on the public key, completing the transaction matching, and ensuring the accurate execution of transactions.

Step 8: The verification of the transaction serial number, which is aimed at preventing replay attacks and tampering.

Step 9: The reputation audit, where the reputation of the transaction initiator is reviewed to ensure compliance with standards, and the quality of the acquired migration dataset is analyzed to issue warnings for datasets with a high suspicion of malicious activity.

Step 10: The generation of zero-knowledge proofs, which protect the privacy of transactions while ensuring their authenticity and validity.

Step 11: The first layer of the blockchain verifies the zero-knowledge proofs to ensure the authenticity and legality of transactions.

Throughout the process, if any step fails to meet the conditions or encounters errors, the flow redirects to the "return error" phase, where error information is reported to facilitate timely issue identification and resolution. Ultimately, when all steps are successfully completed, the first layer of the blockchain will update the newly generated Merkle root.

## Blind privacy data

Participants should register with Bio-Rollup and obtain a key certificate. The critical certificate provides the basis for auditing and account tracing. Figure 2 numbered 1 shows the participant registration process. Figure 4 depicts the data owner registration process.

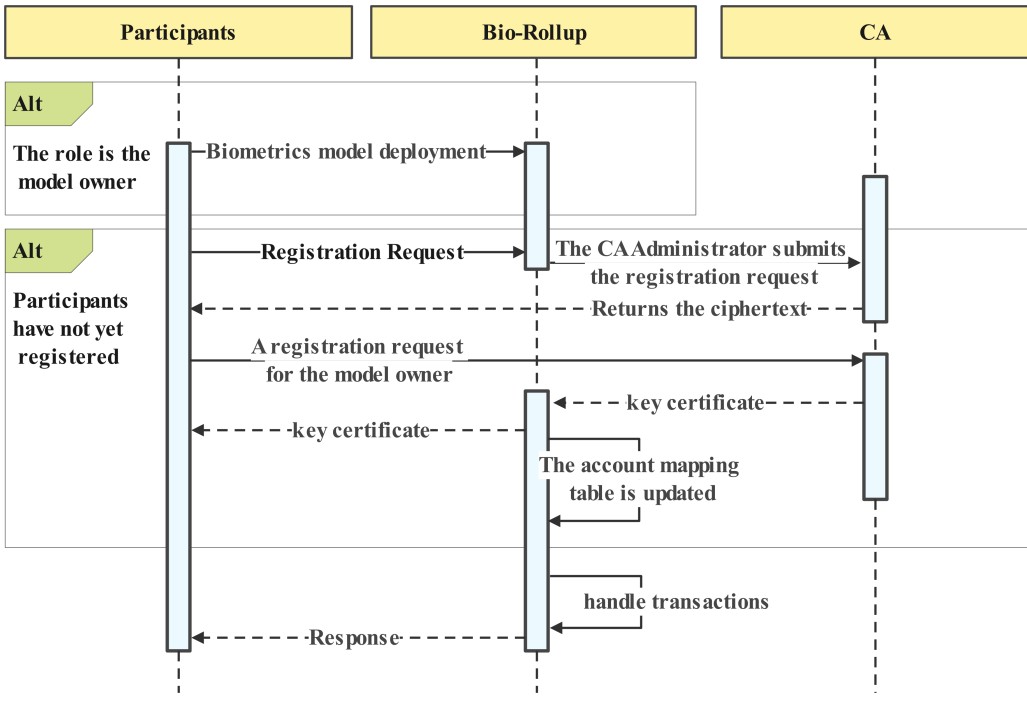

**Figure 4 Participants account initiation.**

The registration process for regular users is similar to that of the data owner. The registration process involves three steps:

Step 1: Unregistered participants request account registration from the system, which then uses the CA administrator account to submit a registration request for the corresponding account name and random cipher text to the CA. The CA processes the request and returns the cipher text.

Step 2: Participants register using the cipher text, generating their signing private key and certificate locally for interaction with the biometric recognition system and blockchain. This step allows users to sign transactions and authenticate their identity on the blockchain.

Step 3: Participants and the system establish a trusted channel and upload their biometric data. The registration of biometric templates will be submitted as a request to the layer two scaling service as an initial transaction for the account. This step enables users to store their biometric information on the blockchain, providing a secure and tamper-proof way to authenticate their identity in Fig. 4.

To ensure the ethics and legality of biometric recognition models and prevent unauthorized tampering, both the deployment and upgrading processes of these models are subject to rigorous audit supervision. This auditing mechanism guarantees the integrity and credibility of the models, thereby safeguarding the secure application of biometric recognition technology. The ownership of the models is unambiguously vested in the data owners, and their deployment is considered the underlying transaction. This setup protects the rights and interests of the data owners and ensures the compliance of model usage.

During the model updating process, the auditing mechanism plays a pivotal role. Any unauthorized attempts to update the models will be promptly detected and blocked by the system, thus ensuring the security and stability of the models.

We used model summaries to better understand the security and credibility of the deployed biometric recognition models. In this process, a hashing function H(·), selected by the data owners, was employed to generate the summaries of the models through hashing operations, $N$ as the serial number of transactions for model employment and updates and $P_k$ as the public key of data owner. These summaries can be utilized for rapid model verification and comparison, serving as a crucial basis for the security and integrity of the models. The registered model summaries are as follows:

$$D_p = \text{SHA256}(\text{H}(\text{Bytes}(model)) : N : P_k). \tag{1}$$

Common users only need to receive summaries of their biometric template classification results. These summaries will be stored as data and provided to Bio-Rollup for auditing. The auditing process aims to gain insight into the user's query intent by thoroughly analyzing their querying behavior. Given that most privacy attacks are based on querying behavior, it is crucial to construct a high-quality transfer dataset, which requires attackers to collect enough diverse data. In biometric recognition systems, misclassification is inevitable due to external factors such as the environment, lighting, and angle, which may lead to inconsistencies in query results for the same user. Therefore, relying solely on inconsistent query results to determine the existence of targeted attacks is not accurate enough. However, from a statistical perspective, this auditing process is still important. The registered biometric template summaries are as follows:

$$D_p = \text{SHA256}(\text{Argmax}(\text{Model}(biometic\ template)) : N : P_k). \tag{2}$$

In the Bio-Rollup framework, the core mechanism of transaction is primarily constructed based on Eqs. (1) and (2). These transactions can be further categorized into two major types: model operations and system queries. Both categories are subject to rigorous auditing supervision within the Bio-Rollup architecture. Notably, the exposure of participant data, including model data and biometric template information, is limited solely to the interaction between the biometric recognition system and the participants. This design ensures that the auditing process cannot access sensitive or private information, thereby effectively protecting the privacy rights of users.

The purpose of auditing extends beyond verifying the legitimacy of models; it also aims to safeguard the compliance of user inquiries. Bio-Rollup provides a secure and reliable transactional environment for participants through this dual safeguard mechanism. Regarding the upload process for the two types of summaries, detailed instructions can be found in Algorithm 1. This algorithm outlines the privacy-preserving approach for data upload and processing, further enhancing the usability and reliability of the Bio-Rollup system.

## Lightweight auditing of transactions

Merkle proofs are a lightweight auditing mechanism based on Merkle trees, enabling users to verify the presence of specific transactions on the blockchain by downloading only the

---

**Algorithm 1** Participants' privacy data upload process.

---

**Require:** Information to be uploaded $m$, Incremental serial number $N$, Participant's private key $S_k$, Participant's metadata $d$.

1: **Procedure** PrivacyDataUpload($m, N, S_k, d$)
2: **if** the participant role in the metadata $d$ is data owner **then**
3:     Generate a model summary $D_p \leftarrow$ SHA256(H(Bytes($model$)) : $N$ : $P_k$)
4: **else**
5:     Generate a template summary $D_p \leftarrow$ SHA256(Argmax(Model($biometic\ template$)))
6: **end if**
7: Serialize data to generate a transaction $TX \leftarrow$ Serialize($[D_p, N, d]$)
8: Sign the transaction $sig \leftarrow$ ECDSA.$sign(TX, S_k)$
9: $TX \leftarrow$ Extend($TX, sig$)
10: Transmit the transaction to the Bio-Rollup node $TLS_{upload}(TX)$
11: **end Procedure**

---

block header and Merkle proof rather than the entire block data. In a blockchain, each block contains a Merkle tree where the leaf nodes are the hashes of all transactions within the block, and the root hash is included in the block header. Users obtain the transaction's hash to be verified, request a full node to generate a Merkle proof, and then calculate the Merkle root using the proof's sibling hashes and path, comparing it with the root hash in the block header to confirm the transaction's existence. This method significantly reduces the data users need to download and store, improving verification efficiency, especially in large-scale blockchain networks. The scalability and efficiency of Merkle proofs make them an ideal choice for lightweight clients to validate transaction existence. Lightweight audit process see Algorithm 2.

Bio-Rollup maintains an account mapping table $M$ on the blockchain's second layer, with each account information occupying 120 bytes. The fields include *Credit* (uint64), *Index* (uint64), *Nonce* (uint64), *Data* (fr.Element), and *PubKey* (ecdsa.PublicKey). These fields collectively represent the complete account information, with *Credit*, *Index*, *Nonce*, and *Data* storing the account's state information and *PubKey* used to verify the account owner's identity. After serialization, the account information forms a binary Merkle tree. In the Bio-Rollup second-layer scaling solution, transactions are manifested in the legal changes of user states, with storage costs linearly increasing with the number of accounts. Zero-knowledge proof technology ensures the validity of transactions and audits, and historical transactions are not stored, making the second-layer storage cost independent of the number of transactions. The design of this solution is lightweight and includes audit mechanisms based on Merkle proofs and storage based on account states.

In the Bio-Rollup architecture, the Operator object assumes the pivotal roles of account information storage and behavior auditing. It incorporates multiple fields and functions tailored for account state management, transaction processing, and the detection of potential attacks. The primary fields of the Operator object include: *InitState* is a byte array for recording the initial state; *State* is a byte array for the current state, including account index, nonce, balance, and public key; *HashState* is a byte array for the hash

---

**Algorithm 2** Lightweight auditing of transactions

---

**Require:** Participant's Transaction $TX$, Participant account mapping $M$.

**Ensure:** Zero-knowledge proof for blockchain layer one verification $\pi_{zk}, \tau_{zk}$, the Attack rate parameter $\xi$.

1: **Procedure** TransactionsAudit($TX, M$)
2:   **for** $i \leftarrow 0$ $to$ $B$ **do**
3:     Retrieve user data from the Bio-Rollup mapping $user \leftarrow$ ReadAccount($M[P_k]$)
4:     **if** the account does not exist **then**
5:       **return** an error indicating that the account does not exist.
6:     **end if**
7:     Obtain merkle proof before state update $\pi_p, \tau_p \leftarrow$ GetWitness($user$)
8:     **if** MerkleProofVerify($\pi_p, \tau_p$) $= False$ **then**
9:       **return** an error indicating that the current state is illegal.
10:    **end if**
11:    Calculate the participant's credit $C \leftarrow \sum_{i=1}^{N}\{(D_p \ \& \ TX_i.D_p) + \log(i+1)[(D_p \ \& \ TX_i.D_p) - 1]\}$
12:    The participant is suspected of strong attack intent if $\xi > 0.5$.
13:    Obtain merkle proof after state update $\pi_n, \tau_n \leftarrow$ SetWitness($user$)
14:    $\pi_{zk}[i], \tau_{zk}[i] \leftarrow (\pi_p, \pi_n), (\tau_p, \tau_n)$
15:   **end for**
16:   **return** $\pi_{zk}, \tau_{zk}, \xi$.
17: **end Procedure**

---

state; *AccountMap* is a field for storing a hash mapping of all available account indices; *nbAccounts* is a field for recording the number of managed accounts; *h* is a field for the hash function used to construct the Merkle tree; *SchemeQueue* is a field for the transaction transmission channel; and *Witnesses* is a field for storing circuit witnesses. This scheme provides the NewOperator function to create an Operator instance, which initializes the state and creates accounts. The readAccount function retrieves account information at a specified index from the Operator instance. The UpdateState function updates the account state of the Operator instance. The AttackDetect function detects possible attack behaviors by calculating the rate of change in account credit limits to identify anomalies. These functions aim to establish an Operator instance capable of efficiently handling account state updates and attack detection.

Upon receiving a transaction, the Bio-Rollup node follows a rigorous verification and auditing process to ensure the security and compliance of the transaction, shown in Fig. 5. This process consists of four core steps, as illustrated in points 3.1 and 3.2 of Fig. 2. The following provides a detailed explanation of these four steps:

Step 1: This step aims to verify the authenticity and validity of the transaction originator. Cross-checking identity identification information such as digital signatures and public keys ensures that only legitimate users can initiate transaction requests.

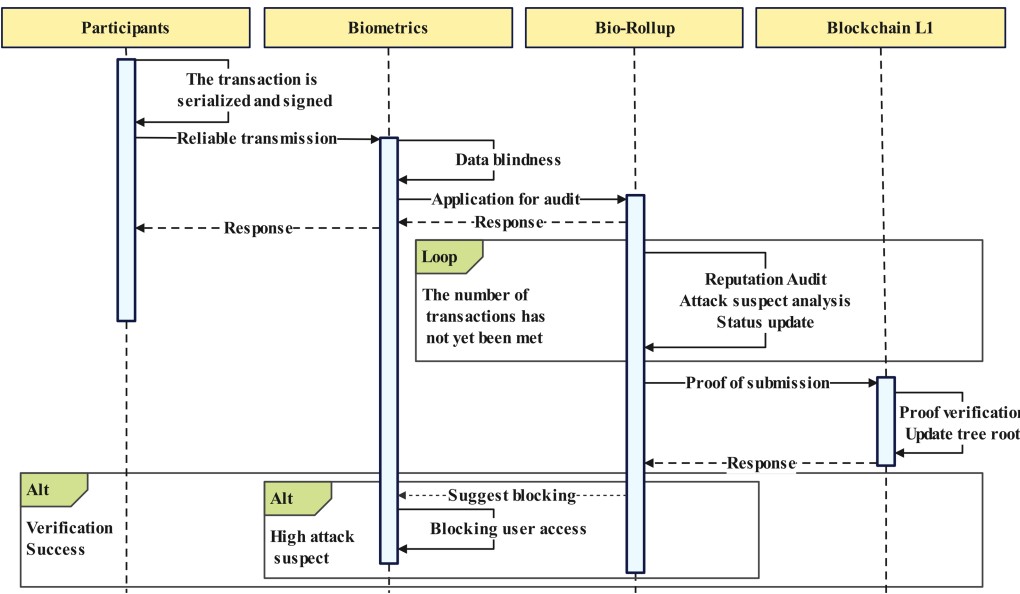

**Figure 5  Data transmission and verification.**

Step 2: Before the transaction enters the verification phase, the Bio-Rollup node examines the legitimacy of the current Bio-Rollup state. This includes checking whether the Bio-Rollup state satisfies preset rules and constraints and identifying potential conflicts or inconsistencies.

Step 3: The node performs detailed transaction validation. This includes, but is not limited to, checking the validity, compliance, and authenticity of the transaction, as well as the authenticity and integrity of the data and information involved in the transaction.

An attacker may utilize a dataset with varying legal sample proportions t after a valid query to query the biometric recognition system. The Nonce serves as the sequence number of the user's request, and the function describes the user credit change with an increase in request numbers, as outlined in Eq. (3):

$$C = \sum_{i=1}^{N} \{ (D_p \,\&\, TX_i.D_p) + \log(i+1)[(D_p \,\&\, TX_i.D_p) - 1] \}. \tag{3}$$

The attack rate parameter $\xi$ represents the likelihood of an attacker exhibiting malicious behavior, and the calculation method can be found in Eq. (4):

$$\xi = 1 - \frac{C}{N}. \tag{4}$$

Step 4: Once the transaction passes verification, the Bio-Rollup node updates the Bio-Rollup state to reflect the transaction's outcome. This update process follows strict rules and standards to ensure that any changes to the state do not introduce any inconsistencies or conflicts.

Once the batch size reaches the standard BatchSize $B$, all publicly and privately disclosed parameters are recorded in a circuit. A zero-knowledge proof corresponding to the protocol

is then generated from the circuit, with the computational cost proportional to the size of the circuit. Bio-Rollup achieves transaction validation and user data settlement separation, offloading a significant amount of computing and storage costs to an off-chain solution. This effectively alleviates pressure on the blockchain. The specific process can be found in Algorithm 3.

---

**Algorithm 3** Zero-knowlege proof verification procedure.

---

**Require:** Zero-Knowledge Proof Provided by Bio-Rollup $\pi_{zk}, \tau_{zk}$, Verification Key Provided by Blockchain Layer one $V_{zk}$.

**Ensure:** Verification result of zero-Knowledge proof.

1:   **Procedure** ZeroKnowledgeProofVerify$(\pi_{zk}, \tau_{zk}, V_{zk})$
2:   Obtain verification result $R \leftarrow$ Verify$(V_{zk}, \pi_{zk}, \tau_{zk})$
3:   **if** R reflects failed verification **then**
4:     **return** Return an error indicating that the proof failed verification.
5:   **end if**
6:   UpdateStateOnLegder$(\tau_{zk})$
7:   **return** $R$
8:   **end Procedure**

---

# EXPERIMENTAL EVALUATION AND DISCUSSION

## Experimental setup

The study employs the CIFAR-10 dataset, a standard benchmark for image recognition algorithms. CIFAR-10 comprises $32 \times 32$ pixel, 3-channel images across ten classes, each containing 6,000 images. The dataset consists of 50,000 training and 10,000 testing images. The deep learning model for the biometric recognition system is based on the pre-trained VGG16 model on the ImageNet-1000 dataset, achieving a top-1 training accuracy of 96% and a top-1 test accuracy of 87% on the CIFAR-10 dataset. Table 4 shows the training results under balance datasets in six neural network architectures, as a benchmark for subsequent experiments. The system is deployed on the Bio-Rollup layer two scaling node using Apache-TVM. The blockchain layer one is implemented as a Hyperledger Fabric consortium chain on Docker, while layer 2 employs the Bio-Rollup scheme developed in this study for auditing user queries. All experiments are conducted on a server equipped with an AMD Ryzen 7 5800H processor, a Radeon graphics processor, 32GB of memory, and an NVIDIA GeForce RTX 3050Ti GPU.

## Dataset balance analysis

Table S7 presents the training results for other imbalanced datasets with different proportions. We used the generalized additive model (GAM) to delve into the critical factors affecting model performance. GAM, an extension of the generalized linear model, incorporates a linear predictor term that is the sum of smooth functions of covariates. The

**Table 4  Training accuracy of the CIFAR-10 dataset under a balanced data set.**

| Percent | Architectures | 200 | 500 | 1000 | 5000 | 10000 | 20000 |
|---|---|---|---|---|---|---|---|
| 0.1 | VGG16 | 23.3% | 32.0% | 49.3% | 71.7% | 75.8% | 80.1% |
| | Resnet18 | 37.7% | 50.8% | 57.1% | 70.7% | 73.5% | 78.4% |
| | Resnet50 | 38.6% | 47.2% | 57.1% | 72.8% | 75.3% | 80.2% |
| | Resnet101 | 33.2% | 44.9% | 57.4% | 70.0% | 71.6% | 77.8% |
| | Densenet121 | 45.7% | 54.0% | 62.7% | 76.4% | 78.5% | 82.5% |
| | Densenet161 | 43.3% | 52.7% | 61.3% | 74.8% | 78.7% | 82.2% |

GAM is specified as:

$$g(\mu_i) = X_i^* \theta + \sum_j f_j(x_{ij}) + \varepsilon, \ \varepsilon \sim N(0, \sigma^2). \tag{5}$$

Here, $\mu_i \equiv E(Y_i)$ is the expected value of the response variable, $Y_i$, which is assumed to follow an exponential family distribution. $X_i^*$ denotes a row of the incidence matrix for parametric model components, $\theta$ is the corresponding parameter vector, $f_j$ represents the jth smooth function of the covariate $x_{ij}$, and $\varepsilon$ is the error term with a normal distribution. To reduce computational costs, smooth functions are often estimated using the reduced rank smoothing approach.

Preliminary exploration suggested that the distribution of $Y_i$ for MPMs was similar to the normal distribution. Consequently, we chose a GAM with a Linear regression and a "identity" link function for model fitting and assessment.

The CIFAR-10 dataset was utilized to construct a generalized additive model (GAM). The model parameters were configured as follows:

- Six artificial deep neural networks were selected, comprising VGG16, Resnet18, Resnet50, Resnet101, Densenet121, and Densenet161, with the number of network layers serving as parameters across six levels: [16, 18, 50, 101, 121, 161].
- The training set size was categorized into six levels: [200, 500, 1,000, 5,000, 10,000, 20,000].
- The proportion of the training set classified as user template output by the model was set at five levels: [0.1, 0.6, 0.7, 0.8, 0.9].

In this experiment, the CIFAR-10 dataset was employed to construct a GAM. The model parameters were configured as follows: the GAM was chosen for data analysis due to its capability to handle continuous, discrete, and categorical data with flexibility. The model's complexity was effectively controlled, and the risk of overfitting was reduced using regularized smoothing parameters, while maintaining strong interpretability by clearly illustrating the impact of each predictor variable.

A summary of GAM is shown in Table 5. The GAM utilized a normal distribution as the link function, conforming to standard statistical assumptions and accurately capturing the data's characteristics. The model's effective degrees of freedom were substantial, reaching 11.5074, indicating the extensive use of degrees of freedom to create a complex model structure that could delve into nonlinear effects. The log-likelihood ratio, AIC, and AICc

**Table 5    The summary of the GAM model.**

|  | Distribution | NormalDist | AIC | 42253.8316 |
|---|---|---|---|---|
| LinearGAM model metrics | Link Function | IdentityLink | AICc | 42255.861 |
|  | Effective DoF | 11.5074 | GCV | 0.0043 |
|  | Log Likelihood | −21114.4084 | Scale | 0.0038 |
|  | Number of Samples | 180 | Pseudo R-Squared | 0.9043 |
| Feature Function | Lambda | Rank | EDoF | P>x |
| s(0) | [285.2329] | 20 | 4.2 | 1.60e−03 |
| s(1) | [2.4758] | 20 | 5.5 | 1.11e−16 |
| s(2) | [8.9639] | 20 | 1.8 | 1.11e−16 |
| intercept |  | 1 | 0.0 | 1.11e−16 |

values were all low, suggesting that the GAM achieved a good balance between data fitting and the avoidance of overfitting. The generalized cross-validation (GCV) value of 0.0043 and the scale value of 0.0038 indicated that the model appropriately incorporated nonlinear components without overfitting or underfitting, and closely fit the data. The pseudo R-squared value of 0.9043 demonstrated that the model could explain most of the variability in the data, confirming the suitability of the GAM. The significance test results for the feature functions and the intercept were highly significant, further highlighting the GAM's superior fitting performance and predictive power.

The fitting results of the GAM on the CIFAR-10 dataset indicate that the training accuracy is significantly influenced by the neural network architecture, training set size, and training set proportion. As demonstrated by the GridSearch method predictions in Fig. 6C, while the model depth increases from 16 to 161, a tenfold expansion, the improvement in accuracy is not substantial. Conversely, the model's accuracy is significantly enhanced with an increase in training set size, consistent with prior research. Moreover, the model's accuracy decreases as dataset imbalance increases, as shown in the Fig. 6C. Considering that typical users tend to query data related to themselves, such as registered biometric templates, rather than other data, the impact of imbalanced datasets on model training accuracy is minimal. Therefore, by analyzing the balance of users' dataset acquisitions, one can assess their query preferences and dataset quality.

## Performance comparison analysis

The batch standard for generating zero-knowledge proofs is $B = [10, 25, 50, 75, 100]$. Simulated users query the biometric recognition system at an interval of 20 ms. Table 6 shows that the proof computation volume of the prover increases with the size of the circuit. Although the volume of the Groth16 proof computation is related to the public input volume, the number of public inputs in the circuit is significantly less than the number of secret inputs, which can be disregarded. Therefore, the Groth16 protocol's proof computation volume depends on the scale of multiplication and connection gates in the circuit, while the PlonK protocol's proof computation volume is related to the scale of multiplication and addition gates. In summary, both protocols' proof computation

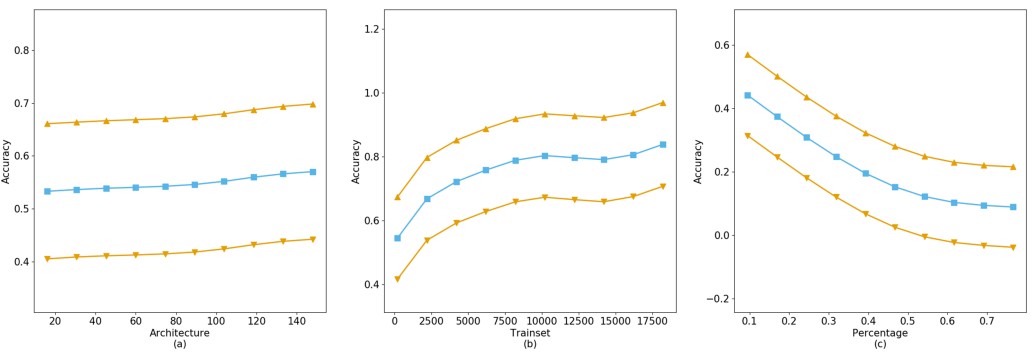

**Figure 6   The accuracy gain of independent parameters on model training.**

**Table 6   Computing scale of zero knowledge proof protocols.**

| Protocol | Proving computing | Verifying computing |
|---|---|---|
| Groth16 | $19770 * B\,E_1, 4942 * B\,E_2$ | $3P, 2 * B\,E_1$ |
| PlonK | $66375 * B * E_1$ | $2P, 18E_1$ |

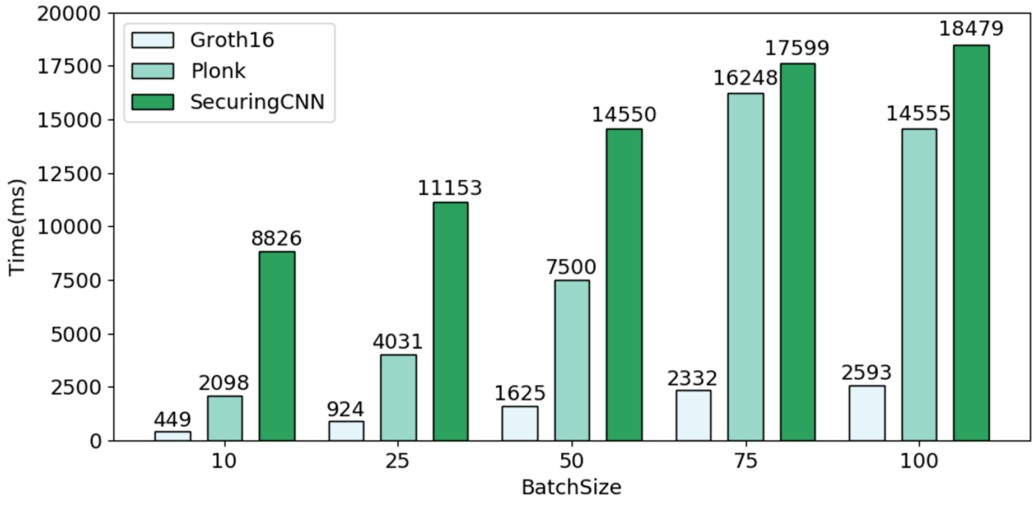

**Figure 7   Comparison of efficiency between Bio-Rollup and SecuringCNN.**

volumes will increase with the size of the circuit, so placing the proof computation process in a second-layer scaling service can reduce the blockchain's computational burden.

As depicted in Fig. 7, through a comparative experiment of overall efficiency among different $B$, the average efficiency of Bio-Rollup based on Groth16 reaches approximately five times that of PlonK protocol and about 11 times that of SecuringCNN scheme. Furthermore, even if Bio-Rollup is implemented using the PlonK protocol suitable for

multi-circuit operations, its average efficiency remains approximately 2.2 times that of SecuringCNN scheme.

The operational expenses of Bio-Rollup are derived from the generation of zero-knowledge proofs by the second-layer scaling service and their verification by on-chain smart contracts. Equations (6) and (7) illustrates the cost $T$ of SecuringCNN and Bio-Rollup:

$$T_{Sec\ CNN} = T_{Feature\ Extracting} + T_{Tamplate\ Matching} \tag{6}$$

$$T_{Bio\ Rollup} = T_{Proving} + T_{Verifying}. \tag{7}$$

In comparison to SecuringCNN, Bio-Rollup does not disrupt the deployment of biometric recognition systems. Rather, it enhances communication with second-layer scaling services through an additional interface. Blockchain prioritizes external system access, input data, and behavior decisions through CA authorization, offline auditing, and zero-knowledge proofs rather than relying on complex feature extraction processes to ensure computing legality. This approach enables blockchain to concentrate on achieving consensus.

## Privacy attacks blocking experiment and analysis

Current privacy attacks are predominantly query-based, necessitating the continuous generation of input for the victim model in the sample space. These inputs, when combined with the victim model's outputs, form a transfer dataset used to train a surrogate model or analyze the data for privacy theft. However, legitimate users typically do not generate many queries that significantly deviate from registered classification labels, which can serve as a basis for determining whether a user is suspected of engaging in privacy attacks. Data set imbalance experiments indicate that when $t$ exceeds 0.5, the parameter does not contribute to model training, hence setting the suspected attack threshold at $\xi = 0.5$ is appropriate.

The horizontal axis of Fig. 8 represents the number of system queries made by the user, denoted as $N$. The vertical axis represents $\xi$, and the curve depicts how $\xi$ changes with the number of user queries under different legal sample proportions $t$. $\alpha$ represents the number of legitimate queries the corresponding account makes to the system before launching an attack.

Setting $\alpha = 500$ (Fig. 8A), the probability of an attacker conducting a suspected attack increases with the proportion of fabricated data. At 588 visits, the dataset with a proportion of $t$ =10% was blocked by Bio-Rollup (solid represents system visits that have been implemented, while dashed represents suspected attacks that were intended but blocked due to lack of intent). At this point, the suspected attackers obtained approximately 90 and 70 pieces of confidence data (*i.e.,* unauthorized access) respectively. The dataset with a proportion of $t$ =50% was blocked by Bio-Rollup at 619 visits. The dataset with a proportion of $t$ =70% was blocked by Bio-Rollup at 822 visits. Extensive experiments revealed that only when the AttackRate value of the dataset with a proportion of $t$ =90% finally converged between 0.4 and 0.5 without being blocked did Bio-Rollup successfully block most malicious

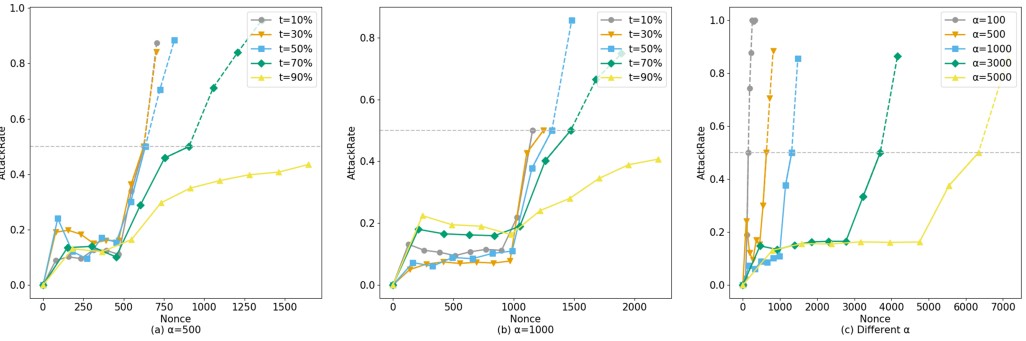

**Figure 8 Comparison of AttackRate under different attack conditions.**

attackers. However, if an attacker with a proportion of $t = 90\%$ visited 5,000 times, they would obtain approximately 450 pieces of confidence data, implying they could obtain about 9% of the confidence data from 5,000 visits. In this scenario, the so-called attacker requires many legitimate samples for cover, and Bio-Rollup can set longer visiting intervals, effectively "transforming" these non-malicious attackers. Setting $\alpha = 1,000$, as shown in Fig. 8B, the situation is similar to $\alpha = 500$ and is not further elaborated. Additionally, attackers may conduct attacks at any time, as shown in Fig. 8C. Setting $t = 50\%$, $B = 10$, starting from privacy attack requests with $\alpha = [100, 500, 1000, 3000, 5000]$, these five conditions were blocked by Bio-Rollup at visits $[144, 619, 1212, 3491, 5868]$ respectively.

Bio-Rollup can successfully block most malicious attackers and transform non-malicious potentially convertible attackers. Considering both the scale of fabricated data used for attacking $\beta$ and the time point for carrying out privacy attacks $\alpha$, we can model $\xi$ based on these two parameters. At this point, the range of $\xi$ is given by Eq. (8):

$$\left[ \frac{\beta + \log \prod_{i=\alpha}^{\alpha+\beta}(i+1)}{N}, \frac{\beta + \log \prod_{i=N-\beta+\alpha}^{N}(i+1)}{N} \right]. \tag{8}$$

Common privacy attacks are analyzed using Eq. (8). In model extraction attacks, the accuracy of attacks against multilayer perceptron(MLP) models depends on the size of the model. $q$ is the number of queries made by the attacker, and $k$ is the number of model parameters. According to *Tramèr et al. (2016)*, as a factor, the average query time can basically achieve the effect of accurately restoring model parameters. That is, the number of queries is half the size of the model parameter. If the attacker can maintain $\xi$ within the minimum value in the range of Eq. (8), to perform 100 million ($\alpha = 100,000,000$) queries for the VGG-16 model with a parameter size of half, an account needs to make approximately 100 million legal queries in advance.

Suppose a potential attacker can maintain $\xi$ at its minimum value $\xi_{\min}$. At this point, considering the data balance analysis, let the size of the dataset Trainset be $s$, then $s = \alpha + \beta$. Let the proportion of user queries classified as their registered biometric template results be $t$, then $\xi_{\min}$ is expressed as a function $\xi(s, t)$ as follows:

$$\xi(s, t) = \frac{\log \Gamma(s+1) - \log \Gamma(st+1) + s(1-t)}{\delta s} \tag{9}$$

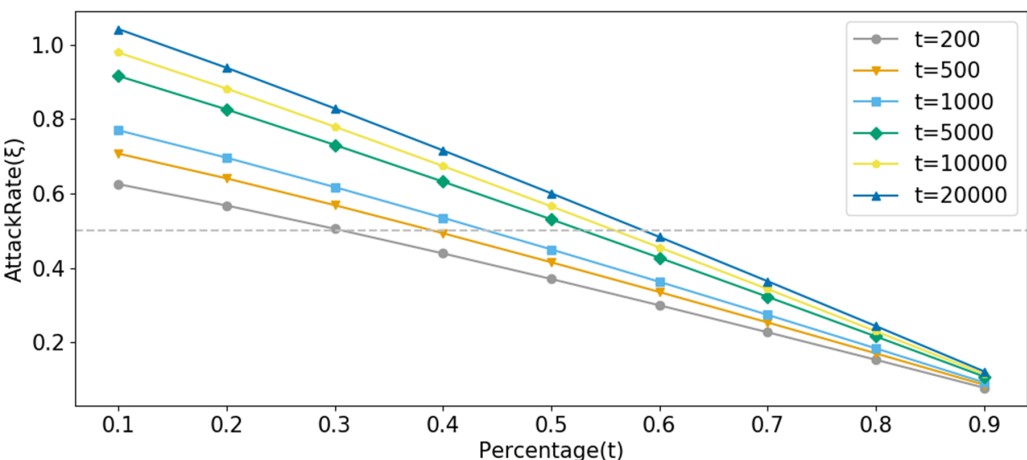

**Figure 9** Different sample sizes for privacy leak monitoring.

where $\Gamma(x)$ is the Gamma function, and $\delta$ ($\delta \geq 1$) is the inverse decay factor, adjusted to control the sensitivity of privacy monitoring. The smaller the value of $\delta$, the higher the sensitivity in monitoring small samples. In this study, $\delta$ is set to $\log(s_{max})$.

As shown in Fig. 9, a threshold parameter $\Xi$ is set to quantify the degree of suspected privacy leakage risk. Within this framework, when $\xi(s, t) > \Xi$, it can be determined that there is a high suspicion of privacy leakage. In Fig. 9, the value of $\Xi$ is set to 0.5 to evaluate the effectiveness of the $\xi(s, t)$ function in privacy leakage monitoring. The function was tested on sample datasets of varying sizes, containing 200, 500, 1000, 5000, and 10,000 samples, respectively. The results show that under the condition of small-scale samples, the value of $\xi(s, t)$ did not trigger the warning mechanism; however, as the sample size increased, especially when the number of samples reached 10,000, the setting of $\Xi = 0.5$ successfully triggered the alert, indicating that the function demonstrates significant efficacy in identifying potential privacy attacks. This finding emphasizes the importance of the reputation decay factor in revealing and preventing privacy leakage, particularly for accounts that have already conducted many query operations and may be subject to account hijacking or privilege abuse. This mechanism allows potential privacy attack behaviors to be effectively identified and controlled, significantly reducing the risk of privacy leakage.

In conclusion, the experiments and analyses conducted above confirm the effectiveness of Bio-Rollup in preventing common privacy attacks. Currently, there is a shortage of blockchain security frameworks for privacy attacks due to the lack of publicly known techniques such as model extraction, member inference, and model inversion. This article proposes a security framework based on a two-layer scalability-focused consortium blockchain that utilizes biometric recognition technology.

## CONCLUSION

This article investigates the application of blockchain technology in biometric recognition systems and introduces an innovative two-tier scaling solution, Bio-Rollup. By integrating

biometric recognition with blockchain, Bio-Rollup effectively decouples the tight integration of biometric recognition and blockchain in existing solutions, significantly enhancing the efficiency of model deployment and upgrades. The solution's lightweight auditing mechanism ensures model integrity while analyzing user query intent to prevent privacy leaks. Moreover, the definition of private data exposure ranges and the separation of the model from the blockchain architecture significantly reduce the risk of model parameter leakage. The application of zero-knowledge proof technology further safeguards participants' sensitive information.

Regarding scalability, transaction volume testing demonstrates Bio-Rollup's excellent scalability in large-scale systems. As the number of transaction-proof batches per 20 ms increases from 10 to 100, the transaction confirmation time increases by only 15%, far below linear growth, validating its robust scalability. Regarding privacy protection, Bio-Rollup implements multiple measures, including advanced encryption technologies, to ensure the secure transmission and storage of sensitive biometric data. The system adheres to the principle of data minimization, collecting only the minimum data necessary to achieve functionality. It emphasizes the importance of user consent, ensuring that users have explicit consent for collecting and processing their biometric data.

Regarding legal compliance, Bio-Rollup complies with global data protection regulations such as GDPR, ensuring that data processing activities meet legal standards and respect the rights of data subjects. Regarding responsibility and ethics, Bio-Rollup follows ethical guidelines such as fairness and transparency. Regarding technical reliability, Bio-Rollup aims to reduce misidentification and denial of service incidents, ensuring the error rate remains acceptable.

In the future, we plan to delve deeper into artificial intelligence's privacy and security risks and improve the Bio-Rollup solution to become a broader privacy protection security framework. We will research universal privacy attack detection and prevention mechanisms for AI, explore more advanced cryptographic techniques, and investigate the application of Bio-Rollup in a broader range of scenarios, such as cross-domain authentication and telemedicine. Additionally, we will promote interoperability between different blockchain platforms, enhance user control over personal data, and automate data management using smart contracts to ensure transparency and compliance. Interdisciplinary research will collaborate with multidisciplinary experts to assess the social and ethical impacts of technology. Research on applying Bio-Rollup in edge computing environments will further reduce data transmission and mitigate privacy risks.

## APPENDIX

Table 7 displays the training accuracy of six artificial neural networks on the Cifar-10 dataset with different proportions of imbalance.

**Table 7 Training accuracy of CIFAR-10 dataset under imbalanced dataset.**

| Percent | Architectures | 200 | 500 | 1,000 | 5,000 | 10,000 | 20,000 |
|---|---|---|---|---|---|---|---|
| 0.6 | VGG16 | 19.5% | 19.6% | 21.2% | 25.3% | 39.3% | 47.9% |
| | Resnet18 | 18.8% | 13.7% | 19.5% | 33.6% | 47.2% | 53.1% |
| | Resnet50 | 14.9% | 18.1% | 16.8% | 34.1% | 43.7% | 54.3% |
| | Resnet101 | 14.6% | 17.9% | 21.0% | 38.4% | 45.2% | 52.5% |
| | Densenet121 | 16.1% | 20.3% | 18.2% | 37.8% | 47.7% | 55.6% |
| | Densenet161 | 18.8% | 15.1% | 25.3% | 43.2% | 49.3% | 56.1% |
| 0.7 | VGG16 | 15.4% | 18.8% | 20.2% | 33.7% | 31.3% | 32.0% |
| | Resnet18 | 14.5% | 17.6% | 23.3% | 33.4% | 39.8% | 51.8% |
| | Resnet50 | 14.8% | 14.2% | 18.3% | 35.3% | 41.9% | 50.7% |
| | Resnet101 | 16.5% | 18.0% | 16.4% | 32.9% | 42.7% | 49.5% |
| | Densenet121 | 17.3% | 19.4% | 21.3% | 37.8% | 46.2% | 54.6% |
| | Densenet161 | 16.3% | 17.5% | 26.4% | 37.5% | 44.4% | 52.5% |
| 0.8 | VGG16 | 16.9% | 16.5% | 17.4% | 26.9% | 29.3% | 38.5% |
| | Resnet18 | 10.7% | 16.1% | 19.9% | 28.6% | 40.0% | 44.6% |
| | Resnet50 | 10.9% | 17.5% | 18.2% | 25.8% | 35.5% | 44.6% |
| | Resnet101 | 11.2% | 14.6% | 20.1% | 25.5% | 37.9% | 44.4% |
| | Densenet121 | 12.3% | 18.5% | 18.6% | 34.9% | 40.1% | 47.5% |
| | Densenet161 | 10.6% | 14.5% | 21.2% | 30.3% | 38.0% | 49.0% |
| 0.9 | VGG16 | 14.8% | 16.1% | 14.4% | 22.4% | 31.0% | 32.8% |
| | Resnet18 | 11.4% | 11.1% | 15.7% | 23.2% | 27.7% | 35.3% |
| | Resnet50 | 10.2% | 18.2% | 17.1% | 19.3% | 22.9% | 31.8% |
| | Resnet101 | 12.8% | 11.6% | 18.1% | 21.1% | 24.1% | 31.5% |
| | Densenet121 | 15.6% | 11.0% | 15.9% | 23.5% | 29.2% | 33.6% |
| | Densenet161 | 12.0% | 14.7% | 16.9% | 22.6% | 27.8% | 36.2% |

## Funding

This work was supported by the Excellent program of Chinese higher education pedagogy society (GJXHSZSZZY023). The funders had no role in study design, data collection and analysis, decision to publish, or preparation of the manuscript.

## Grant Disclosures

The following grant information was disclosed by the authors:
The Excellent program of Chinese higher education pedagogy society: GJXHSZSZZY023.

## Competing Interests

The authors declare there are no competing interests.

## Author Contributions

- Jian Yun conceived and designed the experiments, prepared figures and/or tables, and approved the final draft.

- Yusheng Lu performed the experiments, performed the computation work, prepared figures and/or tables, and approved the final draft.
- Xinyang Liu analyzed the data, authored or reviewed drafts of the article, formal analysis, Resources, writing—review and editing, and approved the final draft.
- Jingdan Guan analyzed the data, authored or reviewed drafts of the article, visualization, Validation, Data curation, and approved the final draft.

## Data Availability

Code and data are available at Zenodo:

Jian, Y. (2024). Bio-Rollup: a new privacy protection solution for biometrics based on two-layer scalability-focused blockchain. Zenodo. https://doi.org/10.5281/zenodo.11596201.

## Supplemental Information

Supplemental information for this article can be found online at http://dx.doi.org/10.7717/peerj-cs.2268#supplemental-information.

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
