# Peer review of "Bio-Rollup: a new privacy protection solution for biometrics based on two-layer scalability-focused blockchain"

_PeerJ Computer Science, doi:10.7717/peerj-cs.2268_

## Round 0.1 · original submission · Major Revisions

We regret to inform you that your manuscript has not been accepted for publication in its current form. More details are needed. The novelty is not clear. Proof of concept should be discussed in detail.

**Language Note:** The review process has identified that the English language must be improved. PeerJ can provide language editing services - please contact us at [email protected] for pricing (be sure to provide your manuscript number and title). Alternatively, you should make your own arrangements to improve the language quality and provide details in your response letter. – PeerJ Staff

Reviewer 1 ·

Basic reporting

Clarity and Structure: Ensure that the paper is clearly written and logically structured. Each section should seamlessly lead to the next. Specifically, the transition between the problem statement, the proposed solution, and the discussion of results could be made smoother to enhance reader understanding.

Literature Review: Expand the literature review to include a broader range of existing solutions, their limitations, and how Bio-Rollup addresses these gaps. This will help in positioning your work more firmly within the current state of research. You can add few related work such as: Privacy preserving authentication system based on non-interactive zero knowledge proof suitable for Internet of Things

Technical Details: Provide more detailed explanations of the technical mechanisms underlying Bio-Rollup, especially the two-layer scaling solution and the lightweight auditing mechanism. A deeper dive into how these components work will aid in comprehensibility.

Experimental design

Experimental Setup: Describe the experimental setup in greater detail, including the datasets used, the configuration of the blockchain environment, and the specifics of the biometric recognition systems tested. This information is crucial for replicability.

Comparison Baseline: Ensure that the baseline models or systems used for comparison are appropriate and represent the current state-of-the-art. It would be beneficial to compare Bio-Rollup against a wider variety of existing solutions to demonstrate its effectiveness more comprehensively.

Robustness Testing: Include tests for the robustness of Bio-Rollup under different conditions, such as varying network loads, attack vectors, and data volumes. This will provide a more rounded evaluation of its performance.

Validity of the findings

Statistical Analysis: Where possible, incorporate statistical analysis to strengthen the validity of your findings. This could include confidence intervals, significance tests, or other relevant metrics that support the conclusions drawn from your experimental results.

Limitations and Assumptions: Clearly state any limitations of your study and the assumptions made during the experimental design. Discussing these openly will provide a more balanced view of the research and its applicability.

Scalability and Generalizability: Address the scalability of Bio-Rollup and its applicability to different blockchain and biometric systems. Providing evidence or discussion on these aspects would enhance the paper's relevance to a broader audience.

Additional comments

Ethical Considerations: Delve deeper into the ethical implications of biometric recognition systems, especially in relation to privacy and data protection. Discuss how Bio-Rollup aligns with ethical guidelines and legal standards.

Future Work: Outline potential areas for future research, including any planned improvements to Bio-Rollup or new features to be developed. This could also include exploration into the integration of additional biometric modalities or blockchain technologies.

Practical Applications: Provide more examples of practical applications and real-world scenarios where Bio-Rollup could be deployed. This will help readers understand its potential impact and the contexts in which it can be most beneficial.

Reviewer 2 ·

Basic reporting

More analysis is important to convince the readers - please go through the more related literatures.

Experimental design

For the quality of the paper, it would be better to compare it quantitatively and qualitatively with other architectures.

Validity of the findings

Before concluding, it would be better to evaluate the study conducted under the discussion and its situation to other studies and the following studies.

Reviewer 3 ·

Basic reporting

In this paper, the authors proposed biometric recognition and blockchain-based two-layer scaling solution. To protect sensitive information of participants, zero-knowledge proof is used. Implementation results show that the proposed solutions provide much better running times compared to literature solutions.

-The abstract is not well written. The authors should rewrite it by giving basic details about the method and novelty of the paper.
-The introduction part needs to be rewritten since it is not well-structured. It should explain the importance of the privacy protection solution with biometrics and blockchain.
-The author provides information on AI-related privacy protection approaches in the first part of the manuscript. However, this does not give the current state of the art.
-What will be the main contribution of proposing a privacy protection solution based on biometric recognition and blockchain. It should be clarified.
-The organization of the introduction section is not well defined. The authors explained some concepts in that part and gave some literature. They should build a connection between the ideas. By making the connection, they should feature the importance of the proposed idea.
-What is the main contribution of this paper to literature? By proposing this scheme, what open problem is solved? Clarify them. The authors should also add some parts to highlight the main motivation of the paper.
-The language is terrible. It needs proofreading.
-The originality and novelty of proposed method is not up to date.
-The related work section does not cover the current state of the art. By systematically analyzing the existing literature, the authors should update the related work section.
-A notation table should be added and security-related definitions should be given in another subsection.
-The formal definition of the proposed scheme should be given with the protocol flow structure to make understanding more transparent. In other words, the proposed method should be explained by providing an algorithm, pseudocode, or something like that.
-The entire workflow of the scheme should be detailed by adding a step-by-step explanation.

Experimental design

-The authors should add a new section to discuss the literature. Then, they also need to explain the main differences of the proposed idea from the literature by making a comparison with previous studies.
In the experiment analysis section is not meaningful.
-The author should add a new section to present the detail of analyzed privacy attacks.
-They should provide detailed evidence that the proposed scheme is efficient for privacy protection requirement by giving mathematical proof and experimental analysis.
-The abstract and conclusion do not describe the content/scope of the paper. The methods expressed in the paper should be summarized comprehensively. The advantages and shortcomings should be briefly summarized.
-Figures 5-7 make no sense for practical use cases.

Validity of the findings

-There is no theoretical evidence to ensure the lightweight auditing of transactions.
-The correctness of the proposed scheme is unclear. Firstly, the authors should define how a privacy protection technique can be verified. Then, by looking at the proposed idea, they should write the correctness analysis.

Additional comments

-In my opinion, the most critical shortcoming of the study is the lack of meaningful comparison results, detailed explanation of the proposed idea, and limited contribution.


Overall. I recommend for rejection.

---

## Round 0.2 · accepted · Accept

We are happy to inform you that your manuscript has been accepted for publication since the reviewers' comments have been addressed.

Reviewer 1 ·

Basic reporting

Authors updated the paper as per my previous comments.

Experimental design

As above

Validity of the findings

As above

Reviewer 2 ·

Basic reporting

The authors have addressed comments and concerns in their revised manuscript.

Experimental design

The authors have addressed comments and concerns in their revised manuscript.

Validity of the findings

The authors have addressed comments and concerns in their revised manuscript.